# The Impact of a Time Gap on the Process of Building a Sustainable Relationship between Employee and Customer Satisfaction

**Lukasz Skowron [1,*]**, **Marcin Gąsior [1]** and **Monika Sak-Skowron [2]**

[1] Department of Marketing, Faculty of Management, Lublin University of Technology, 20618 Lublin, Poland; m.gasior@pollub.pl

[2] Department of Enterprise Management, Faculty of Social Sciences, Catholic University of Lublin, 20950 Lublin, Poland; monika.sak-skowron@kul.pl

[*] Correspondence: l.skowron@pollub.pl; Tel.: +48-723696452

**Abstract:** The aim of this paper is to describe the relationships between changes in employee indices (motivation and satisfaction) and customer indices (satisfaction and loyalty) in a single- and multi-term perspective. The article presents the results of primary research conducted in two industries (banking services and shopping centers) during three annual reference periods. The authors used the PLS-SEM method in the analytical process. The results of the research suggest that there is a strong relationship between changes in the areas of employee and customer satisfaction in the studied sectors, with a one-year time shift, which the authors called the "time gap". In addition, it turned out that the strength of influence of the employee's motivation level on customers is clearly lower than the strength of influence of the employee satisfaction. The occurrence of a "time gap" between employee and customer processes suggests that any changes introduced in the area of customer service as well as broadly understood human resource management policy need some time to become sustainable—to be noticed by the market and coded in the minds of the recipients of the offer as the new and currently applicable standard. The article makes a successful attempt at a long-term analysis of the relationship between employees and customers, assuming a time delay between both phenomena. As a result of the conducted research, it was possible to operationalize the discussed relationship in terms of strength and direction as well as the time shift.

**Keywords:** customer satisfaction; customer loyalty; employee motivation; employee satisfaction; PLS-SEM

## 1. Introduction

The subject of the present article is the analysis of the strength and direction of relationships linking employee processes (described by the level of employee motivation and satisfaction index) and customer processes (expressed by the level of customer satisfaction and loyalty index). In the research, the authors managed to demonstrate that, between the two main market actors, i.e., customers purchasing all sorts of goods on the one hand and employees of enterprises and institutions producing and supplying these goods to the market on the other, what is taking place is a single, more or less sustainable, cause-and-effect macro-process. Although the analyzed relationship between the employee and the customer seems to be quite logical and intuitive, not much empirical proof exists in the scientific literature.

In order to be able to analyze the discussed relationship, it was necessary to collect empirical data from several consecutive reference periods in both environments. Such a database made it possible to present the relationships occurring between the two studied phenomena not only in the same reference

period but also taking into account the one- or two-year delay occurring between measurements of the indices of both examined groups of stakeholders, i.e., employees and customers.

As annual research, the authors collected opinions of customers and employees of a group of enterprises operating in one of two market sectors—large-scale shopping centers and banks. In the analytical process, the results of measuring the customer processes of a given unit were compared with the corresponding measurement results of its employees. The analysis of both processes was carried out for the results obtained in the same period of time and taking into account the "time gap"—time shift of one and two years between the "earlier" process, i.e., the one concerning the employee, and the "subsequent" one regarding the customer.

The discussed research is, in general, related to the relationship between employee behavior and its impact on customers' (delayed) reactions. From such a perspective, it is directly connected with two dimensions of sustainability—social and economic. Assuming a customer-centric sustainability, as defined by Sheth et al. [1], the social dimension is related to the impact of consumption on personal wellbeing and the economic dimension on economic wellbeing. Customer satisfaction and, in general, a positive relation with a company may and will affect both areas, e.g., by reducing stress, creating positive social interactions and relationships, enabling more well-considered purchases, and thus reducing both costs for consumers and impact on the environment.

From the companies' point of view, understanding the character of the relationship between employees and customers, especially in terms of time between stimuli and reactions, may be even more important. The assumption that a market's reaction to possible changes in employee motivation or his or her satisfaction is not immediate should lead to a reduction in additional actions aimed at employee (e.g., training and reorganizations but also layoffs) or directly at customers (e.g., marketing), therefore saving resources and labor and, possibly, creating a better work environment and enabling sustainable development of an organization.

In the article, the authors first concentrate on a literature review concerning three main areas: modeling of employee motivation and satisfaction, modeling of customer satisfaction, and the relationship between abovementioned phenomena. Then, the explanation and description of the research subject, research period, and sampling procedure has been given. Afterwards, the authors in detail explain the process of the construction of the SEM models used in the research. Results of the research, which are the next part of the paper, are presented in the tables as well as in the graphical form of the two holistic models of relations between employee and customer processes for both analyzed industries (Figure 1—shopping centers and Figure 2—banks). The article is summarized by the conclusions, discussion, and the research limitations.

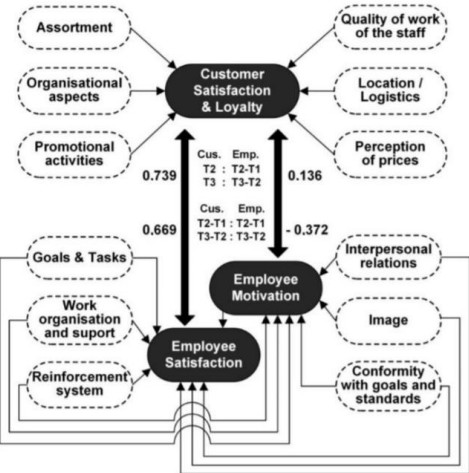

**Figure 1.** Holistic model of relations between employee and customer processes—the shopping center sector.

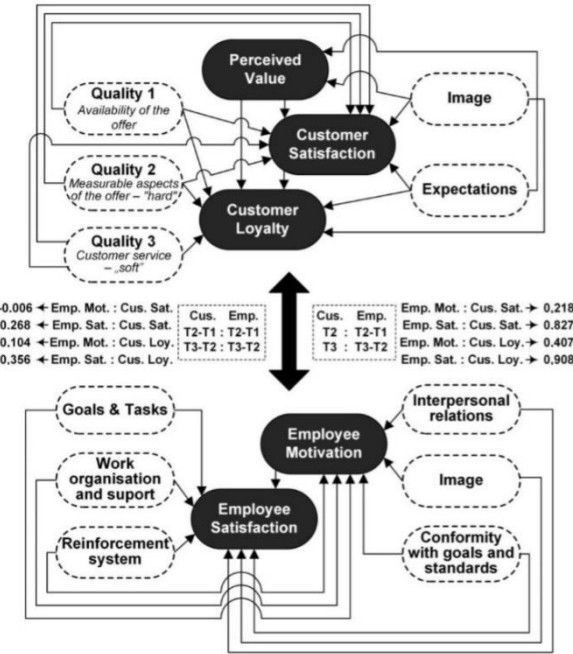

**Figure 2.** Holistic model of relations between employee and customer processes—the banking sector.

## 1.1. Models of Employee Motivation and Satisfaction

Theories explaining the sources and mechanisms of the formation of motivation can be divided into those based on the content, i.e., focused around needs and the feeling of deficiency associated with them, those based on the process, striving to explain why people choose specific ways of satisfying their set of needs, and those looking at motivation from the perspective of reinforcements, seeking to explain why some behaviors are continued and maintained, while others are not [2].

Among content theories, particularly noteworthy is the Self-Determination Theory, in which the needs and sources of motivation are arranged on a continuum—from motivation based on full autonomy to motivation forced by external factors [3]—and include the need to have competences, autonomy, and social contacts [4,5]. Among the theories of motivation based on the concept of the process, the Goal-Setting Theory [6] can be pointed out. It is the basis for creating a model combining the goals, effort, satisfaction, and commitment of the employee, illustrating the high- performance cycle, which has be treated by the authors of the article as the basic framework for the modeling of the employee level occurrences. Employee satisfaction, commitment, and performance are three terms inextricably linked to both a person's place in the organization, their work environment, as well as motivation processes and factors. In the literature, employee satisfaction is mostly understood as a set of attitudes towards their work and work environment [7,8]. The basis for creating commitment is the individual's attachment to the organization, including compliance and internalization. The sources of potential commitment should be sought in the characteristics of the organization and the personality of the individual and their professional experience, while organizational commitment itself can be manifested in three forms: affective, continuance, and normative commitment [9].

The relationship between employee satisfaction and motivation is also found in motivation theories, where the source of satisfaction lies in experiencing intrinsic motivation, which also affects the level of organizational commitment [10]. Employee satisfaction, due to its potential impact on the effectiveness of the individual and organization, but also on the formation of both positive and negative attitudes, has in itself become the subject of numerous studies and modeling approaches. The first is Lent and Brown's model [11], resulting from the previously developed Social Cognitive Career Theory. It assumes the existence of four areas shaping employee satisfaction (participation and progress in the implementation of tasks and adopted goals, conditions and results of the performed work, expectations

regarding one's own effectiveness and ability to perform duties, as well as personality traits and emotional state—see [12]). The next model has proven the relationship between employee effectiveness and their work satisfaction, whose strength can be modified by the level of effort associated with task performance, remuneration and participation in profit [3,10]. The employee's personality is the key variable that has a significant impact on the mechanisms of creating satisfaction. Additionally, Trull and Widiger presented a reviewof the studies on the relationship between the employee's personality and their job satisfaction [12].

Studies of sources of employee satisfaction indicate that the level of employee satisfaction additionally depends on the form of employment, including flexibility of work, autonomy and independence [13], non-business activities of the enterprise [14], the relationship between one's remuneration and that of other employees [15], calling toward work [16] as well as the employee's state of health [17,18]. What is more, according to some research results, the most vital motivational factors are those connected with the social and financial aspects [19]. However, there is also research whose results indicate that, in some regions (for example, Asia), most non-financial benefits neither motivate nor increase job satisfaction [20].

### 1.2. Customer Satisfaction Models

In defining the concept of customer satisfaction, three dimensions can be distinguished: psychological (satisfaction is the customer's experience, integrating elements of emotional reactions with cognitive processes), relativistic (comparing the subjective experiences and feelings of the buyer with the adopted reference point—expectations, experiences, competitive offers, opinions of environments), and empirical (refers to consumption experiences or subsequent stages of the product use and purchasing process—see [21,22]).

Regardless of many different definitions of the concept of satisfaction, two main approaches may be defined [23,24]: transaction-specific satisfaction and cumulative satisfaction. Both aforementioned definitions of satisfaction have been treated by the authors as equally important basic points of reference in their further research. What is more, shaping the phenomenon of satisfaction should be treated not as a linear process but as a combination of quantitative and qualitative aspects of a given offer and the previous potential of customer market experience. In addition to this, in the literature, one can find much research which supports the findings of the key role of different after-sales services elements in the customer satisfaction building process [25,26].

The complexity of the mechanism of shaping customer satisfaction finds expression in many models explaining the essence of the analyzed process [27,28]. The following models should be pointed out:

- the emotional model;
- the equity theory model (transaction satisfaction model);
- the expectancy disconfirmation model;
- the models of service quality and the quality gap model;
- the concept of creating satisfaction based on the PROSAT model.

When it comes to modeling of the abovementioned phenomena of customer satisfaction, path models occupy a special place both in the theory and practice due to their dual nature. On the one hand, they are a theoretical construct with which the origins of the discussed phenomenon can be explained; on the other, they are used as measuring tools which extensively employ the structural equation modeling (SEM) methodology [29]. There are several basic problems and difficulties associated with measuring satisfaction:

- satisfaction changes over time: new experiences and a growing level of awareness change the potential level of satisfaction that can be achieved,
- satisfaction is generally complex and constitutes a mix of experiences that occur before, during, and after it was measured,

- satisfaction is born in social contexts that are diverse and changeable,
- the reasons for satisfaction can be difficult to express, especially where less perceptible aspects of services are being considered,
- the reasons for dissatisfaction may be easier to express, especially if it is an exceptional situation,
- without understanding the reasons for satisfaction, there is a danger that a "good result" can be treated as a reason not to change anything, seeing it mainly as a PR tool.

A correctly planned and conducted customer satisfaction survey should fulfill three basic interrelated functions: the corrective, learning, and stimulus functions.

The authors in their research decided to use the PLS-SEM modeling approach (SEM with use of the partial least squares regression statistical method—more detailed information in the Materials and Methods paragraph) as the main means of analysis of both employee and customer phenomena.

### 1.3. The Relationship between the Level of Employee Motivation and Satisfaction and the Level of Customer Satisfaction

The relationship between the level of employee motivation and satisfaction and the level of customer satisfaction and their potential loyalty seems to be a logical and rational element of the relationship that links the company with its environment; therefore, it has been at the center of interest of researchers and business practitioners for many years. It is worth emphasizing, however, that these relationships are not entirely clear-cut, and the studies described in the literature prove their various nature and scope, which is influenced by the type of business, the specificity of the industry, and the moderating factors adopted in the assumptions of the conducted analyses.

A very interesting attempt to interpret the discussed relationship is presented in the meta-analysis of data obtained from 42 research projects carried out in 36 companies and institutions (commercial facilities, banks, insurance companies, schools, food service establishments). The research focused mainly on the search for relationships between employee satisfaction and their commitment and consumer satisfaction, as well as profit and productivity of the analyzed market entities, staff turnover, and the number of accidents [30]. It showed the existence of a relatively moderate (though statistically significant) correlation between employee satisfaction and commitment and between customer satisfaction and loyalty (average Pearson correlation coefficients were 0.32 and 0.33, respectively), as well as enterprise productivity (0.20 and 0.25). The obtained results do not directly indicate the direction of the observed relations, although the authors themselves, referring to the works by J. H. Fleming, assume that it is employee satisfaction and commitment that are the source of the observed effects [31].

Very similar conclusions result from the study on the relationship between employee satisfaction, customer satisfaction, and quality of services [32]. The research sample amounted to a total of 6680 respondents. The obtained results showed a weighted average correlation between the analyzed areas of satisfaction (of employees and customers) at the level of 0.25 and between employee satisfaction and the quality of services amounting to 0.29. In addition, it was shown that the quality of service is a mediating factor (the so-called moderator) between employee satisfaction and customer satisfaction.

H. Jeon and B. Choi, who chose the educational services market as their area of research [33], were also among the authors involved in the analysis of employee–customer relations (from the perspective of satisfaction of both groups of stakeholders). In the employee model, the authors included areas such as employee roles in the organization, role conflicts, job satisfaction, commitment, and willingness to continue working in the current place. The customer model, on the other hand, included quality of interactions, quality of the result of the service process, customer satisfaction, trust, and loyalty. A statistically significant relationship between employee satisfaction and customer satisfaction was confirmed. In addition, research showed that the analyzed relationship is mainly the result of two variables that characterize the employees themselves: their confidence in themselves and their skills, as well as their attitudes towards cooperation with others. Along with the intensification of these

employee characteristics, the impact of the level of satisfaction experienced by the employee on the level of satisfaction felt by the customer also increases.

The first researchers who drew attention in their works to the two-way form of relations between employee and customer processes were Salanova, Agut, and Peiró. The authors focused on analyzing the relationship between customer loyalty and conditions of providing services (service climate), understood as an employee's observations concerning practices, procedures, and behaviors regarding customer service and the quality of services offered which are expected in the organization, as well as propagated and awarded by it. The adopted research model assumed that the conditions of providing services are affected by the organization's resources (e.g., technology used, level of employee autonomy and training, interpersonal relations within the organization). In this case, customer loyalty was shaped by the subjectively perceived employee effectiveness and, as the results of the conducted research showed, also indirectly affected the previously presented conditions of providing the service [34].

The relationship between employee satisfaction and customer satisfaction has also been the subject of much research in the sectors of services [35–38]. Generally, the obtained results allowed the researchers to confirm two main research hypotheses:

- There is a direct positive correlation between employee satisfaction and customer satisfaction. The authors of the study explained this fact by a higher level of motivation found in satisfied employees, who were therefore more willing to provide high-quality services.
- The relationship between employee satisfaction and the financial results of an entity is not direct—the moderating factor in the case of the results of the discussed research is customer satisfaction.

In addition to the review of the results of research on the analyzed relationship, additional significant conclusions should be emphasized [39]:

- the impact of organizational resources on the conditions of providing services, and thus on the perceived effectiveness of employees and customer loyalty, is not direct but is expressed by shaping employee commitment;
- customer loyalty is perceived by employees as positive feedback, further improving the conditions of providing services.

All cited research proves the existence of a real relationship between the level of employee motivation and satisfaction and customer satisfaction and loyalty. However, the adopted methodology and the limited time range of the research (it was not repeated in the subsequent reference periods) makes it difficult to unambiguously determine both the direction of the analyzed relationships and the phenomenon of the "time gap" occurring between the observed changes in the area of employee and customer processes.

## 2. Materials and Methods

### 2.1. The Research Subject, the Research Period, and a Description of the Research Sample

The subject of the study was to identify processes that shape the satisfaction and motivation of company employees and the satisfaction and loyalty of their customers (as external stakeholders), including, in particular, an analysis of all factors determining these processes and of the relationships occurring between these processes (referred to here as employee and customer processes). To achieve this research goal, the authors adopted a holistic research model, involving measurement of both employee and customer processes, carried out within the same business entities, repeated three times at annual intervals.

The research was carried out in the years 2013–2016. It was conducted among employees and customers of both groups of entities: large-scale shopping centers and bank offices operating in a major Polish city. The choice of these units was supported by the fact that, firstly, they represent two forms

of activity in which the employee has direct contact with the customer—that is, trade and services. Secondly, these are large entities employing a large number of staff and, at the same time, are used by a wide range of customers. Although the dataset may be considered quite old now, the analysis is aimed at the diagnosis of general characteristics of the relations occurring between researched employee and customer satisfaction (from the perspective of 3-year annual time period) rather than the specific industry or market-related issues. Certainly, the picture of the market, companies, and their customers evolves, but we assume that the postulated relationship is stable and time-independent in terms of its existence and general shape. In this sense, the gap between the time of data collection and the article submission do not affect the overall research results and findings associated with the importance and the character of the "time gap" phenomenon occurring between the observed changes in the area of analyzed employee and customer processes.

For the purposes of the research, three primary path relation models were developed to measure the following:

- motivation and satisfaction of employees of the Lublin banking sector and of shopping centers (one common model was developed for employees of both sectors);
- satisfaction and loyalty of the customers of the Lublin banking sector;
- satisfaction and loyalty of the customers of the Lublin shopping center sector.

The analysis of research results was based on the PLS-SEM method, which allowed the authors to present a synthetic index expressing the level of employee satisfaction and motivation as well as customer satisfaction and loyalty and to identify and measure the determinants that shape them at each of the stages of the research, as well as to present an analysis of the trends in the changes in the discussed indices and the direction and strength of the mutual relations between them.

In the case of the study of the sector of shopping centers, the authors narrowed the scope of observations to the three largest entities that functioned in all reference periods (hereinafter referred to as SC1, SC2, and SC3—in the years covered by the survey, the three largest shopping centers selected were indicated as the most visited ones by over 90% of the research sample).

In the banking sector, on the other hand, data were obtained from the two largest banks (referred to as Bank 1 and Bank 2), which, according to data from 2014, serve over 37% of the market for personal current accounts in Poland (data according to the Office of the Polish Financial Supervision Authority, 2014.)

During the surveys of the shopping centers' customers, conducted in subsequent, annual reference periods, respectively 587, 698, and 512 valid questionnaires were acquired and accepted for analysis, while in the surveys of customers of the banking sector, 480, 366, and 340 were acquired (the sample was distributed relatively evenly between the studied entities in both analyzed sectors). Since it had been impossible to define and obtain any rational sampling frame for selected entities, the authors decided to use quota sampling. The participants were selected in such a way that the obtained quotas represented the structure of the general population, from the point of view of two key criteria: gender and age (data according to the Central Statistical Office in Poland, 2014–2016.)

In the case of groups of employees, due to the smaller size of the surveyed population, all accessible population members were invited to participate. The obtained samples were slightly smaller and amounted respectively to 208, 172, and 179 persons for the shopping center sector and 174, 164, and 150 persons for the banking sector.

## 2.2. Construction of Research Models

### 2.2.1. Model for Measuring Employee Motivation and Satisfaction

The model of measuring the sources and level of employee motivation and satisfaction was developed mainly on the basis of the European Employee Index [40]. This model assumes that the level of employee satisfaction and motivation is shaped by such factors as the image of the organization,

evaluation of the work of the management, evaluation of the work of superiors, interpersonal relations and cooperation, working conditions, remuneration and opportunities for self-development. Basing on these assumptions and additionally taking into account the observations and conclusions resulting both from other popular methods of measuring the level of employee motivation and satisfaction (i.e., Minnesota Satisfaction Questionnaire and Satisfaction Survey, SERVQUAL) and from the results of scientific research conducted on individual components of the discussed model construction [5,40–44], the authors developed their own original measurement model, which is presented in Figures 1 and 2. It takes into consideration six areas grouping independent variables and two areas grouping dependent variables: goals and tasks (5 variables); interpersonal relations in the entity (15 variables); evaluation of the reinforcement system (9 variables); work organization and support of the entity (5 variables); the image of the subject (3 variables); conformity of work with one's own goals and standards (6 variables), and employee motivation and satisfaction (respectively, 6 and 4 variables).

After developing the path model, an analysis was carried out of its basic measures of internal stability (AVE—average variance extracted, Cronbach's alpha, and composite reliability) and of external stability in relation to the obtained empirical material ($R^2$). The obtained values are presented in Tables 1 and 2, respectively, for the shopping center sector and the banking sector. Taking into account the obtained internal stability indicators, it can be assumed that the developed model is correct, and the analyzed variables were adequately assigned to individual research areas. In the case of the measure of external stability ($R^2$), expressing the degree to which the proposed model explains the variance in the data set, the obtained result should be considered very good—for most of the studied subjects and measurement periods, the value of $R^2$ exceeds 0.65, even reaching 0.80. This attests to a very good fitting of the model to the obtained empirical data, thus proving its correctness and validating its use for further analyses of the considered phenomena.

**Table 1.** Measures of stability of the employee processes model—the shopping center sector.

| Areas of the Model | Average Variance Extracted (AVE) | | | Cronbach'sAlpha | | | Composite Reliability | | | $R^2$ | | |
|---|---|---|---|---|---|---|---|---|---|---|---|---|
| | 2013 | 2014 | 2015 | 2013 | 2014 | 2015 | 2013 | 2014 | 2015 | 2013 | 2014 | 2015 |
| **Shopping Center (SC) 1** | | | | | | | | | | | | |
| Goals and tasks | 0.61 | 0.68 | 0.56 | 0.84 | 0.89 | 0.80 | 0.88 | 0.91 | 0.86 | | | |
| Interpersonal relations | 0.69 | 0.44 | 0.64 | 0.97 | 0.91 | 0.96 | 0.97 | 0.92 | 0.96 | | | |
| Image | 0.67 | 0.72 | 0.65 | 0.76 | 0.81 | 0.72 | 0.86 | 0.89 | 0.85 | | | |
| Organization and support | 0.74 | 0.60 | 0.61 | 0.91 | 0.83 | 0.83 | 0.93 | 0.88 | 0.88 | | | |
| Reinforcements | 0.60 | 0.51 | 0.43 | 0.91 | 0.88 | 0.82 | 0.93 | 0.90 | 0.87 | | | |
| Conformity with goals and standards | 0.55 | 0.51 | 0.64 | 0.83 | 0.81 | 0.88 | 0.88 | 0.86 | 0.91 | | | |
| Motivation | 0.56 | 0.60 | 0.52 | 0.82 | 0.87 | 0.81 | 0.88 | 0.90 | 0.86 | 0.79 | 0.62 | 0.85 |
| Satisfaction | 0.79 | 0.70 | 0.65 | 0.86 | 0.79 | 0.73 | 0.92 | 0.88 | 0.84 | 0.73 | 0.82 | 0.75 |
| **Shopping Center (SC) 2** | | | | | | | | | | | | |
| Goals and tasks | 0.58 | 0.72 | 0.69 | 0.82 | 0.92 | 0.92 | 0.87 | 0.93 | 0.90 | | | |
| Interpersonal relations | 0.47 | 0.52 | 0.60 | 0.91 | 0.93 | 0.90 | 0.92 | 0.94 | 0.92 | | | |
| Image | 0.77 | 0.89 | 0.79 | 0.84 | 0.94 | 0.89 | 0.91 | 0.96 | 0.92 | | | |
| Organization and support | 0.69 | 0.54 | 0.61 | 0.89 | 0.81 | 0.86 | 0.92 | 0.85 | 0.89 | | | |
| Reinforcements | 0.59 | 0.53 | 0.58 | 0.91 | 0.90 | 0.86 | 0.93 | 0.91 | 0.89 | | | |
| Conformity with goals and standards | 0.56 | 0.45 | 0.61 | 0.84 | 0.65 | 0.85 | 0.88 | 0.76 | 0.90 | | | |
| Motivation | 0.50 | 0.44 | 0.55 | 0.80 | 0.73 | 0.75 | 0.85 | 0.82 | 0.82 | 0.74 | 0.50 | 0.68 |
| Satisfaction | 0.73 | 0.44 | 0.60 | 0.82 | 0.56 | 0.77 | 0.89 | 0.70 | 0.81 | 0.74 | 0.59 | 0.72 |
| **Shopping Center (SC) 3** | | | | | | | | | | | | |
| Goals and tasks | 0.62 | 0.59 | 0.60 | 0.85 | 0.82 | 0.85 | 0.89 | 0.88 | 0.90 | | | |
| Interpersonal relations | 0.53 | 0.51 | 0.56 | 0.93 | 0.93 | 0.89 | 0.94 | 0.93 | 0.88 | | | |
| Image | 0.53 | 0.53 | 0.51 | 0.53 | 0.54 | 0.62 | 0.76 | 0.76 | 0.73 | | | |
| Organization and support | 0.76 | 0.72 | 0.69 | 0.92 | 0.90 | 0.90 | 0.94 | 0.93 | 0.91 | | | |
| Reinforcements | 0.59 | 0.59 | 0.55 | 0.91 | 0.91 | 0.91 | 0.93 | 0.93 | 0.93 | | | |
| Conformity with goals and standards | 0.62 | 0.50 | 0.64 | 0.87 | 0.78 | 0.82 | 0.91 | 0.85 | 0.92 | | | |
| Motivation | 0.59 | 0.42 | 0.57 | 0.86 | 0.71 | 0.73 | 0.90 | 0.79 | 0.84 | 0.69 | 0.65 | 0.69 |
| Satisfaction | 0.77 | 0.69 | 0.78 | 0.85 | 0.77 | 0.80 | 0.91 | 0.87 | 0.89 | 0.71 | 0.66 | 0.70 |

**Table 2.** Measures of stability of the employee processes model—the banking sector.

| | Areas of the Model | Average Variance Extracted (AVE) | | | Cronbach'sAlpha | | | Composite Reliability | | | R² | | |
|---|---|---|---|---|---|---|---|---|---|---|---|---|---|
| | | 2013 | 2014 | 2015 | 2013 | 2014 | 2015 | 2013 | 2014 | 2015 | 2013 | 2014 | 2015 |
| **Bank 1** | Goals and tasks | 0.44 | 0.80 | 0.76 | 0.68 | 0.94 | 0.92 | 0.78 | 0.95 | 0.91 | | | |
| | Interpersonal relations | 0.54 | 0.50 | 0.54 | 0.93 | 0.93 | 0.77 | 0.94 | 0.93 | 0.71 | | | |
| | Image | 0.75 | 0.91 | 0.71 | 0.84 | 0.95 | 0.91 | 0.90 | 0.97 | 0.88 | | | |
| | Organization and support | 0.74 | 0.25 | 0.61 | 0.91 | 0.87 | 0.72 | 0.93 | 0.49 | 0.69 | | | |
| | Reinforcements | 0.44 | 0.68 | 0.65 | 0.85 | 0.94 | 0.93 | 0.87 | 0.95 | 0.89 | | | |
| | Conformity with goals and standards | 0.63 | 0.53 | 0.61 | 0.88 | 0.81 | 0.84 | 0.91 | 0.87 | 0.81 | | | |
| | Motivation | 0.50 | 0.54 | 0.57 | 0.80 | 0.82 | 0.85 | 0.86 | 0.87 | 0.89 | 0.64 | 0.58 | 0.63 |
| | Satisfaction | 0.72 | 0.63 | 0.69 | 0.81 | 0.71 | 0.84 | 0.89 | 0.84 | 0.87 | 0.65 | 0.83 | 0.79 |
| **Bank 2** | Goals and tasks | 0.43 | 0.30 | 0.59 | 0.65 | 0.84 | 0.72 | 0.78 | 0.51 | 0.71 | | | |
| | Interpersonal relations | 0.40 | 0.52 | 0.52 | 0.88 | 0.93 | 0.73 | 0.89 | 0.94 | 0.72 | | | |
| | Image | 0.65 | 0.83 | 0.71 | 0.73 | 0.90 | 0.89 | 0.85 | 0.94 | 0.87 | | | |
| | Organization and support | 0.65 | 0.61 | 0.62 | 0.87 | 0.86 | 0.85 | 0.90 | 0.88 | 0.85 | | | |
| | Reinforcements | 0.55 | 0.56 | 0.52 | 0.89 | 0.90 | 0.85 | 0.91 | 0.92 | 0.81 | | | |
| | Conformity with goals and standards | 0.66 | 0.47 | 0.57 | 0.90 | 0.80 | 0.91 | 0.92 | 0.82 | 0.85 | | | |
| | Motivation | 0.44 | 0.50 | 0.54 | 0.74 | 0.79 | 0.80 | 0.82 | 0.85 | 0.81 | 0.71 | 0.71 | 0.72 |
| | Satisfaction | 0.87 | 0.71 | 0.72 | 0.93 | 0.79 | 0.86 | 0.95 | 0.88 | 0.90 | 0.81 | 0.70 | 0.81 |

## 2.2.2. Model for Measuring Customer Satisfaction and Loyalty in the Banking Sector

In order to develop a path model for measuring the level of satisfaction and loyalty of banks' customers, the authors took the EPSI (European Performance Satisfaction Index) model as the starting point. Then, using the modular nature of the path modeling methodology, the original formula of the research model was expanded by adding 3 areas of quality (instead of two that the EPSI model contains) and expanding the number and scope of variables describing individual latent variables, so that the constructed structure was best suited to the realities of the studied sector and market. When developing the final design of the research model, numerous results of studies on the issues addressed here were taken into account [44–46]. The research model developed by the authors consists of 5 areas grouping independent variables and 3 areas grouping result variables (the graphical presentation of the model is shown in Figure 1). The independent areas include image (described by 6 variables); customer expectations (10 variables); the first area of quality, describing the availability of banking services and products (10 variables); the second area of quality, taking into account the measurable quality of the offer of banking products and services (8 variables), and the third area of quality, expressing the quality of customer service (6 variables). The group of dependent areas includes perceived value (7 variables); customer satisfaction (3 variables), and customer loyalty (3 variables).

In order to verify the correctness of the constructed model, the authors, analogously as in the case of the employee model, conducted an analysis of internal stability (AVE, Cronbach's alpha, and composite reliability indicators) and of external stability (R2) of the proposed framework structure (for both studied banks in each of the three reference periods). The results of the analyses are presented in Table 3.

The analysis of measures of internal stability of the model shows that particular research questions were correctly assigned to specific latent variables of the model. The only exception is the results of the AVE indicator obtained for the area of quality 1 (results in some reference periods are below the adopted limit of 0.5). This situation is a consequence of a wide range of issues addressed within the discussed area. It should be noted, however, that slightly worse results of the AVE indicator are not further confirmed by the results of other indicators of internal stability of the model (Cronbach's alpha and composite reliability), which allows for adopting the assumed framework structure of the path model for the banking sector as the basis for further research and analysis.

When assessing the measure of external stability of the model, the $R^2$ values obtained for the areas of perceived value, satisfaction, and loyalty show that the model provides a reliable picture of the mechanisms shaping the discussed phenomena (the model explains the occurring variances in the cited areas in each case in over 50%, sometimes even reaching a result of over 70%).

**Table 3.** Measures of the stability of the customer processes model—the banking sector.

| Areas of the Model | | Average Variance Extracted (AVE) | | | Cronbach's Alpha | | | Composite Reliability | | | $R^2$ | | |
|---|---|---|---|---|---|---|---|---|---|---|---|---|---|
| | | 2014 | 2015 | 2016 | 2014 | 2015 | 2016 | 2014 | 2015 | 2016 | 2014 | 2015 | 2016 |
| Bank 1 | Expectations | 0.60 | 0.60 | 0.50 | 0.93 | 0.93 | 0.85 | 0.94 | 0.94 | 0.88 | | | |
| | Image | 0.66 | 0.76 | 0.54 | 0.89 | 0.94 | 0.83 | 0.92 | 0.95 | 0.87 | | | |
| | Quality 1 | 0.53 | 0.53 | 0.42 | 0.90 | 0.90 | 0.85 | 0.92 | 0.92 | 0.88 | | | |
| | Quality 2 | 0.56 | 0.65 | 0.54 | 0.89 | 0.92 | 0.88 | 0.91 | 0.94 | 0.91 | | | |
| | Quality 3 | 0.65 | 0.72 | 0.58 | 0.89 | 0.92 | 0.85 | 0.92 | 0.94 | 0.89 | | | |
| | Perceived value | 0.59 | 0.74 | 0.53 | 0.88 | 0.94 | 0.85 | 0.91 | 0.95 | 0.89 | 0.55 | 0.79 | 0.54 |
| | Satisfaction | 0.55 | 0.78 | 0.67 | 0.77 | 0.86 | 0.75 | 0.77 | 0.92 | 0.86 | 0.54 | 0.59 | 0.50 |
| | Loyalty | 0.65 | 0.60 | 0.69 | 0.74 | 0.78 | 0.78 | 0.85 | 0.81 | 0.87 | 0.67 | 0.57 | 0.64 |
| Bank 2 | Expectations | 0.56 | 0.58 | 0.47 | 0.91 | 0.92 | 0.88 | 0.93 | 0.93 | 0.90 | | | |
| | Image | 0.53 | 0.60 | 0.63 | 0.83 | 0.87 | 0.88 | 0.87 | 0.90 | 0.91 | | | |
| | Quality 1 | 0.48 | 0.48 | 0.49 | 0.86 | 0.88 | 0.89 | 0.89 | 0.90 | 0.91 | | | |
| | Quality 2 | 0.50 | 0.60 | 0.54 | 0.86 | 0.90 | 0.88 | 0.89 | 0.92 | 0.90 | | | |
| | Quality 3 | 0.61 | 0.69 | 0.63 | 0.87 | 0.91 | 0.88 | 0.90 | 0.93 | 0.91 | | | |
| | Perceived value | 0.58 | 0.64 | 0.58 | 0.88 | 0.91 | 0.88 | 0.91 | 0.93 | 0.90 | 0.56 | 0.65 | 0.55 |
| | Satisfaction | 0.59 | 0.73 | 0.65 | 0.74 | 0.81 | 0.73 | 0.81 | 0.89 | 0.85 | 0.54 | 0.57 | 0.56 |
| | Loyalty | 0.63 | 0.62 | 0.62 | 0.71 | 0.70 | 0.70 | 0.84 | 0.83 | 0.83 | 0.63 | 0.70 | 0.62 |

### 2.2.3. Model for Measuring Customer Satisfaction and Loyalty in Shopping Centers

As in the case of the banking sector, the starting point in the process of developing a path model describing the phenomenon of customer satisfaction and loyalty for the shopping center sector was also the structure of the EPSI model. In this case, as in the former, the list of standard issues was supplemented with problems specific to the sector in question. As a result of conducted literature research on possible factors determining the satisfaction of shopping center customers [47–49] and an analysis of the results of pilot studies, a set of 6 main independent areas of the model was developed, together with the result area aggregating the issues of customer satisfaction and loyalty (the graphical presentation of the model is shown in Figure 2). (Due to the very high level of correlation between the areas of satisfaction and loyalty (over 0.80 for each reference period), it was decided, despite the obvious differences between the discussed phenomena, to combine areas of customer satisfaction and loyalty into one result meta-area.) Selected areas of the research model cover assortment (13 variables); organizational aspects (15 variables); promotional activities (7 variables); quality of work of the staff (7 variables); location/logistics (8 variables); perception of the price level (9 variables); customer satisfaction and loyalty as one result meta-area (4 variables for satisfaction and 3 for loyalty).

The verification of the correctness of the constructed model was again based on the analysis of internal stability (AVE, Cronbach's alpha, and composite reliability indicators) and of external stability (R2) of the proposed framework structure in relation to the obtained empirical material (see Table 4).

When assessing the measure of external stability, the value of R2 in each case exceeds the reference level of 0.5 (reaching the level of 0.67), which proves the usefulness of the model for mapping the mechanism of formation of the discussed phenomena.

The analysis of internal stability measures of the model shows, however, that in some areas there is a relatively large thematic divergence of the addressed issues (levels of AVE in some areas range from 0.4 to 0.5). (The norm accepted for social research assumes a level not lower than 0.5.) The latent variables with the lowest level of consistency (for all studied entities and reference periods) were assortment, organizational aspects, and location/logistics. In these, the range of included factors represented by manifest variables is very broad.

When assessing the level of internal consistency of the model, it should be emphasized that the authors of the study were fully aware of the difficulties in precisely assigning the examined issues to individual areas of the path model, which resulted from the wide range of services provided by the discussed entities and the multi-attribute process of valuing the market offer by potential customers. However, not wanting to artificially lower the significance of some areas represented by a relatively smaller number of research questions, the authors decided to combine them, creating larger analytical

areas. The justification for this approach is further confirmed by the analysis of other internal stability indicators of the model (Cronbach's alpha and composite reliability), which are without exception within the standards adopted for social research (minimum value of 0.7 for Cronbach's alpha and 0.8 for composite reliability—see, e.g., Vicky, 2006).

To summarize, the obtained values of stability measures of all the presented path models, from the perspective of both the studied sectors and entities as well as all adopted reference periods, make it possible to treat the obtained results of the PLS-SEM analysis as a stable and reliable image of the mechanism of shaping the discussed phenomena (customer satisfaction and loyalty, as well as employee motivation and satisfaction).

**Table 4.** Measures of the stability of the customer processes model—the shopping center sector.

| Areas of the Model | | Average Variance Extracted (AVE) | | | Cronbach'sAlpha | | | Composite Reliability | | | R² | | |
|---|---|---|---|---|---|---|---|---|---|---|---|---|---|
| | | 2013 | 2014 | 2015 | 2013 | 2014 | 2015 | 2013 | 2014 | 2015 | 2013 | 2014 | 2015 |
| Shopping Center (**SC**) 1 | Assortment | 0.50 | 0.50 | 0.48 | 0.87 | 0.87 | 0.87 | 0.90 | 0.89 | 0.89 | | | |
| | Perception of prices | 0.51 | 0.71 | 0.57 | 0.87 | 0.95 | 0.91 | 0.89 | 0.96 | 0.92 | | | |
| | Promotional activities | 0.68 | 0.62 | 0.65 | 0.92 | 0.91 | 0.91 | 0.94 | 0.92 | 0.93 | | | |
| | Organizational aspects | 0.47 | 0.50 | 0.56 | 0.92 | 0.89 | 0.92 | 0.93 | 0.90 | 0.93 | | | |
| | Location/logistics | 0.45 | 0.40 | 0.49 | 0.82 | 0.78 | 0.79 | 0.87 | 0.84 | 0.84 | | | |
| | Quality of service | .67 | 0.61 | 0.62 | 0.92 | 0.89 | 0.90 | 0.93 | 0.91 | 0.92 | | | |
| | Satisfaction and loyalty | 0.59 | 0.53 | 0.51 | 0.82 | 0.78 | 0.84 | 0.87 | 0.84 | 0.88 | 0.57 | 0.58 | 0.57 |
| Shopping Center (**SC**) 2 | Assortment | 0.48 | 0.58 | 0.43 | 0.87 | 0.91 | 0.84 | 0.88 | 0.92 | 0.84 | | | |
| | Perception of prices | 0.52 | 0.57 | 0.46 | 0.88 | 0.91 | 0.80 | 0.91 | 0.92 | 0.60 | | | |
| | Promotional activities | 0.71 | 0.59 | 0.59 | 0.93 | 0.88 | 0.89 | 0.94 | 0.91 | 0.91 | | | |
| | Organizational aspects | 0.46 | 0.44 | 0.43 | 0.92 | 0.89 | 0.87 | 0.93 | 0.88 | 0.88 | | | |
| | Location/logistics | 0.51 | 0.42 | 0.41 | 0.80 | 0.76 | 0.77 | 0.85 | 0.81 | 0.80 | | | |
| | Quality of service | 0.69 | 0.52 | 0.63 | 0.93 | 0.85 | 0.90 | 0.94 | 0.88 | 0.92 | | | |
| | Satisfaction and loyalty | 0.59 | 0.52 | 0.57 | 0.82 | 0.84 | 0.82 | 0.87 | 0.88 | 0.86 | 0.57 | 0.66 | 0.67 |
| Shopping Center (**SC**) 3 | Assortment | 0.52 | 0.50 | 0.52 | 0.88 | 0.92 | 0.88 | 0.90 | 0.93 | 0.90 | | | |
| | Perception of prices | 0.45 | 0.68 | 0.50 | 0.87 | 0.94 | 0.88 | 0.88 | 0.95 | 0.90 | | | |
| | Promotional activities | 0.64 | 0.64 | 0.61 | 0.90 | 0.91 | 0.89 | 0.93 | 0.93 | 0.92 | | | |
| | Organizational aspects | 0.46 | 0.48 | 0.45 | 0.91 | 0.92 | 0.91 | 0.93 | 0.93 | 0.92 | | | |
| | Location/logistics | 0.44 | 0.44 | 0.43 | 0.82 | 0.82 | 0.75 | 0.86 | 0.86 | 0.80 | | | |
| | Quality of service | 0.70 | 0.63 | 0.60 | 0.93 | 0.90 | 0.89 | 0.94 | 0.92 | 0.91 | | | |
| | Satisfaction and loyalty | 0.59 | 0.59 | 0.55 | 0.82 | 0.82 | 0.86 | 0.87 | 0.87 | 0.89 | 0.56 | 0.53 | 0.60 |

## 3. Results

In order to analyze the relationships between the examined areas of employee processes (motivation and satisfaction) and customer processes (satisfaction and loyalty), indices of the discussed result areas in the constructed models were used, which had been determined using the PLS method (see Tables 5 and 6—shopping centers and banks).

**Table 5.** Values of measurement model indices for the shopping center sector.

| Areas of the Model | | Values of Area Indices (for Periods T1, T2, and T3) | | | | | | | | |
|---|---|---|---|---|---|---|---|---|---|
| | | SC 1 | | | SC 2 | | | SC 3 | | |
| | | 2013 | 2014 | 2015 | 2013 | 2014 | 2015 | 2013 | 2014 | 2015 |
| Employee perspective | Goals and tasks | 6.28 | 6.56 | 7.55 | 6.82 | 6.46 | 6.27 | 7.14 | 6.35 | 6.21 |
| | Interpersonal relationships | 7.05 | 7.24 | 8.10 | 7.99 | 7.03 | 6.49 | 7.73 | 7.32 | 7.49 |
| | Image | 6.51 | 6.95 | 6.57 | 6.79 | 7.06 | 7.29 | 7.48 | 6.91 | 7.13 |
| | Organization of work and support | 6.58 | 6.74 | 7.60 | 6.87 | 6.62 | 5.69 | 7.07 | 6.54 | 6.42 |
| | Reinforcements | 6.28 | 6.76 | 7.79 | 6.13 | 7.13 | 7.27 | 6.16 | 6.54 | 6.41 |
| | Conformity with goals and standards | 6.16 | 5.81 | 6.04 | 6.59 | 5.55 | 5.22 | 6.57 | 6.03 | 5.57 |
| | **Motivation** | **6.47** | **6.63** | **7.20** | **7.47** | **6.15** | **5.58** | **7.53** | **6.24** | **5.53** |
| | **Satisfaction** | **5.69** | **6.60** | **6.64** | **6.13** | **6.73** | **6.28** | **6.16** | **5.94** | **5.78** |
| Customer perspective | Assortment | 6.46 | 7.50 | 6.39 | 5.92 | 6.89 | 5.15 | 6.52 | 7.05 | 6.26 |
| | Perception of prices | 5.01 | 6.78 | 5.09 | 5.41 | 6.65 | 5.47 | 4.88 | 6.03 | 4.93 |
| | Promotional activities | 6.37 | 7.17 | 6.14 | 6.14 | 6.96 | 6.30 | 5.96 | 6.84 | 6.02 |
| | Organizational aspects | 7.06 | 7.70 | 7.14 | 6.85 | 7.46 | 7.21 | 7.06 | 7.38 | 7.24 |
| | Location/logistics | 7.79 | 7.80 | 7.93 | 7.69 | 7.66 | 7.88 | 7.12 | 7.43 | 7.22 |
| | Quality of the work of the staff | 7.08 | 7.60 | 7.11 | 6.97 | 7.60 | 7.37 | 6.73 | 7.40 | 6.91 |
| | **Satisfaction and loyalty** | **7.07** | **7.30** | **6.70** | **6.59** | **6.78** | **6.26** | **6.87** | **7.00** | **6.33** |

Table 6. Values of measurement model indices for the banking sector.

| Areas of the Model | | Values of Area Indices (for Periods T1, T2, and T3) | | | | | |
|---|---|---|---|---|---|---|---|
| | | Bank 1 | | | Bank 2 | | |
| | | 2014 | 2015 | 2016 | 2014 | 2015 | 2016 |
| Employee perspective | Goals and tasks | 7.17 | 7.43 | 7.28 | 5.60 | 6.33 | 6.12 |
| | Interpersonal relationships | 7.65 | 7.47 | 7.67 | 7.74 | 7.81 | 7.93 |
| | Image | 7.39 | 7.63 | 7.87 | 6.72 | 7.33 | 7.68 |
| | Organization of work and support | 7.02 | 7.68 | 7.72 | 7.07 | 7.42 | 7.87 |
| | Reinforcements | 5.42 | 6.99 | 7.21 | 6.56 | 6.90 | 7.08 |
| | Conformity with goals and standards | 7.09 | 6.96 | 6.25 | 7.05 | 7.06 | 6.74 |
| | **Motivation** | **8.02** | **6.69** | **5.92** | **6.93** | **6.57** | **6.29** |
| | **Satisfaction** | **7.01** | **7.16** | **7.19** | **6.15** | **6.56** | **7.28** |
| Customer perspective | Expectations | 8.49 | 8.75 | 8.21 | 8.59 | 8.63 | 8.26 |
| | Image | 7.17 | 7.78 | 7.30 | 7.17 | 7.45 | 7.36 |
| | Quality 1 | 7.37 | 7.64 | 7.47 | 7.05 | 7.31 | 7.27 |
| | Quality 2 | 7.34 | 7.73 | 7.65 | 7.14 | 7.55 | 7.84 |
| | Quality 3 | 7.47 | 7.99 | 7.86 | 7.42 | 7.83 | 7.65 |
| | Perceived value | 6.93 | 7.56 | 7.14 | 6.79 | 7.34 | 7.22 |
| | Satisfaction | 6.70 | 7.30 | 7.04 | 6.60 | 7.22 | 7.44 |
| | **Loyalty** | **7.03** | **7.44** | **7.17** | **6.96** | **7.47** | **7.65** |

The first to analyze were the relationships between the discussed dependent variables within a single reference period. Subsequently, relationships were examined, taking into account the annual time shift of the studied customer indices compared to employee indices. It turned out that, from the perspective of both studied sectors, the obtained results did not show any statistically significant correlations between the analyzed employee and customer area indices.

In connection with the above, the authors decided to expand the area of analysis and to examine the relationships between dynamics and directions of changes in the indices of both groups of respondents (employees and customers) both in terms of a single reference period and from the perspective of a one-year shift period (the so-called one-year "time gap"). In order to estimate the dynamics of changes, analyses were carried out for the results expressed both in absolute terms (the number of index points by which the results changed) and in relative terms (the percentage representation of the strength and direction of the observed changes).

The obtained results allowed the authors to observe that, in the case of the shopping center sector, the dynamics of relative changes in the level of the employee satisfaction index have a statistically significant effect on changes in the indices of customer satisfaction and loyalty within one reference period (a correlation of 0.679) and even more clearly when taking into account the one-year time gap (a correlation of 0.837). However, the obtained results show only marginal impact of the discussed changes in motivation of shopping center employees on the analyzed areas from the customer perspective (both taking into account and not taking into account the time gap in the conducted observations—correlations, respectively, of −0.293 and 0.239). Interestingly, the repetition of the same analyses for the banking sector also did not show any statistically significant relationships between directions and the dynamics of changes in employee motivation and satisfaction indices and customer satisfaction and loyalty.

Summarizing the presented research results, it can be observed that the strongest mutual relations between the dependent variables of employee and customer models for both studied industries occur between:

- dynamics of changes in the area of employee satisfaction and levels of result area indices from the customer perspective, determined for the year after the change;
- dynamics and direction of changes occurring within employee areas and dynamics and direction of changes within customer areas observed within the same measurement period.

In Figures 1 and 2, the authors present holistic relational models illustrating in graphical form the discussed results of key dependencies occurring between employee and customer phenomena, separately for the shopping center sector and for the banking sector. In Figure 2, one can see separate statistical data obtained for customer satisfaction and loyalty as well as employee

motivation and satisfaction (consequences of the specific customer model construction for banking services—see Section 2.2.3). In this case, the authors obtained four different path relations results (employee motivation–customer satisfaction; employee satisfaction–customer satisfaction; employee motivation–customer loyalty; employee satisfaction–customer loyalty) for each "time gap" characteristic (T2:T2-T1/T3:T3-T2 time periods and T2-T1:T2-T1/T3-T2:T3-T2 time periods). In the case of Figure 1, the customer model construction for shopping centers combines areas of customer satisfaction and loyalty into one meta-area (see Section 2.2.2) and it results in obtaining only two different path relations results (employee motivation–customer satisfaction and loyalty; employee satisfaction–customer satisfaction and loyalty) for each of the abovementioned "time gap" characteristics.

Data obtained for the shopping center industry, illustrated in Figure 1, show that the strongest relationships within the discussed sector occur between employee satisfaction and customer satisfaction and loyalty:

- the correlation coefficient calculated for the dynamics of change in the indices of both phenomena, reflecting the extent to which a change in one index is accompanied by a change in the other, is 0.669;
- the correlation coefficient expressing the relationship between the dynamics of changes in the employee satisfaction index and the value of the customer satisfaction and loyalty index for the year after the change is 0.739.

However, as it was previously noted in relation to the area of employee motivation in the discussed industry, the obtained levels of correlation coefficient for both examined perspectives (−0.372 and 0.136, respectively) indicate only small mutual influence of the analyzed phenomenon on the area of customer satisfaction and loyalty.

The regularities observed for the shopping center sector are partially confirmed by the results of analyses conducted for the banking sector. Moreover, in this case, it is a clearly stronger relationship, linking the examined customer processes with employee satisfaction, while a weaker relationship links them with their motivation. Moreover, Figure 2 shows that, in the case of the banking sector (as opposed to the shopping center sector), there is greater variation between the obtained correlation coefficients. The results calculated for the relationship between employee satisfaction and customer processes are definitely lower in the case of the relationship between the dynamics of changes in both examined indices and higher in the case of examining the relationship linking the dynamics of change in the employee satisfaction index with levels of customer satisfaction and loyalty indices, determined for a period of one year after the change.

## 4. Discussion and Conclusions

The conducted analyses of relational links between the indices of employee areas (satisfaction and motivation) and customer areas (satisfaction and loyalty), as part of the adopted model constructions, allowed the authors to draw the following conclusions:

1. In the case of both examined sectors, one can observe a much stronger impact of the area of employee satisfaction than their motivation on the results obtained for the customer perspective.
2. The strong and more sustainable relationships found in the case of the "time gap" suggest that one should be very careful with the analysis of market effects of the introduced changes within the broadly understood human resource management policy, because, as the research shows, between the appearance of the stimuli and the occurrence of the expected results on the side of the customer, there is a noticeable time shift.

3. The occurrence of a "time gap" in customer processes suggests that any changes introduced in the area of customer service standards and procedures need some time to be noticed by the market and coded in the minds of the recipients of the offer as the new and currently applicable standard. In the case of regular customers, it is only repeated contact with the new formula of the offer that will allow them to change the way they evaluate a given organization and to treat new standards as the current "reference point", which in turn also leads to a change in the perception of a given company, its products, and services on the market.

Including companies operating on two separate markets—banks and shopping centers—made it also possible to search and evaluate similarities and differences both in processes of creation of motivation, satisfaction, and loyalty, as well as in the relationships connecting employee and customer behavior. The main findings from the comparative analysis of both categories of companies are as follows:

1. For both cases, employee satisfaction has a significantly higher influence on customer satisfaction and loyalty than employee motivation;
2. In the case of banks, one may observe that the relation between employee motivation and customer areas is visibly stronger than the one observed within shopping centers;
3. It was observed that shopping center staff may have a negative influence on customer satisfaction and loyalty when there are overmotivated, presumably by owners or managers, which is proven by the negative correlation between changes in employee motivation and changes in customer satisfaction and loyalty levels.

In the current study, the authors have made a successful attempt to look at both areas in a holistic manner, assuming that these areas are part of one meta-process of building value in the company. In its basis, this process assumes the existence of a real relationship between employees and customers, which, as a result of the analyses carried out by the authors of the study, was operationalized in terms of strength, direction, as well as time shift. Its innovativeness and academic contribution are related to a completely new approach to evaluate the relationship between employee motivation and satisfaction and customer satisfaction and loyalty, which includes the existence of the time shift between them. Such a multi-stage research perspective is unique and currently not explored either by business or scientific research.

Nowadays, both business practitioners as well as scientists seem to agree to a point that the main challenges of 21st century business management will mostly concentrate on nothing else than managing old, well known, intangible assets of the company (such as employee satisfaction and motivation or customer satisfaction and loyalty) by using the new, modern measurement and managing tools and techniques. Presented in the article, the holistic model concept can be treated as a modern and innovative tool which can support the decision making process of the company in the fields of Human Resources Management (HRM) and Customer Relationship Management (CRM). This approach allows managers to look at both organization stakeholders (customers and employees) from the perspective of one process. Thanks to this, one can see how employees' motivation and satisfaction affect customer satisfaction and loyalty (with reference to strength, direction, as well as time shift of this relation), which enables managers to better understand the vital part of the whole value creation process of the company. This knowledge can provide very precise information on how to efficiently and effectively allocate the organization resources in the fields of HRM and CRM. What is even more important is that this holistic PLS-SEM modeling approach gives managers the ability to possess quantitative data of both analyzed processes (which are strictly intangible in their nature), which makes them much easier to control and manage.

Despite the relevance and interesting findings of the presented study, there are several limitations that may diminish its validity. They are mainly related to the obtained sample. Although it is, at least structurally, representative, a random approach would be even more beneficial from the point of view of data quality—unfortunately, such an approach, when it comes to shopping center customers is, at least,

very problematic. As a second main limitation, we assume a long time delay between each stage of measurement. More frequent readouts, although difficult, would provide better data granularity and thus better insight into researched phenomena. There is also the question of whether this kind of relationship, alongside its strength, direction, and the scale of the time shift, would be observed in different service sectors (e.g., car services, healthcare and wellbeing, etc.). Finally, we assume that such research, due to cultural and social factors, may be only valid for Europe. This limitation is often visible in different research related to customer satisfaction and loyalty—it is unknown whether the theory, constructs, measures, and relationships among constructs are applicable to other regions as well [50].

In order to avoid such limitations, we would recommend validating the presented theory on a smaller scale but with a better sample and higher resolution—in research conducted in a particular company, involving more frequent, quarterly measurement and a different approach to data gathering. It could involve, instead of quota (or even random) sampling, conducting the research on a relatively complete set of customers who would make a transaction in some limited period of time along with measuring, at the very same time, employee levels of satisfaction and motivation. Another interesting way of extending and validating the presented findings would be to replicate similar research on a company or companies from different markets. Moreover, for those who would like to implement such a study in different geographical regions, it might be beneficial to evaluate the validity of assumed satisfaction, motivation, and loyalty models.

**Author Contributions:** Conceptualization: L.S., M.G. and M.S.-S.; methodology: L.S. and M.G.; validation: L.S. and M.G.; formal analysis: L.S. and M.G.; investigation: M.S.-S.; data curation: L.S. and M.G.; writing—original draft preparation: L.S., M.G. and M.S.-S.; writing—review and editing: L.S., M.G. and M.S.-S.; visualization: M.S.-S.; project administration: L.S. All authors have read and agreed to the published version of the manuscript.

**Funding:** This research received no external funding.

**Conflicts of Interest:** The authors declare no conflict of interest.

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
