# Peer review of "The Impact of a Time Gap on the Process of Building a Sustainable Relationship between Employee and Customer Satisfaction"

_sustainability, doi:10.3390/su12187446_

Round 1

Reviewer 1 Report

The abstract is formulated in a clear and concise way: the aim of the paper, the research methodology and the results are very well summarized.
The Introduction presents in a structured manner the literature review on the three main elements of the empirical analysis: employee motivation and satisfaction, customer satisfaction and the relation between them. However, personal opinion on these topics should be provided and it should be also explained which are the definitions of the aforementioned concepts used by authors further on in the research (e.g. own definition and delimitation of the concepts on which the present empirical research relies).
It is essential to provide an explanation regarding the time gap between the time period when the research was conducted (2013-2016, as stated in the text) and the submission date of this article. The empirical data is quite old (this is probably the main vulnerability of the article) and it should be emphasized its relevance in the current context.
It is highly appreciated the fact that the authors provide logical explanations for their decisions and research circumstances such as selection of the sample, of the two sectors of activity etc. as well as transparent data about the number if valid responses obtained. The construction of the research model is very well explained, maybe even too many internal details for a scientific article and a mature audience.
The discussion and conclusion part should be extended. The differences between the two sectors should be explained. It should be emphasized the practical relevance of the results for practitioners and managers from both industries and eventually recommendations for HR management and customer care management could be formulated. The research results, if properly explained and formulated could be very useful for companies. If available, further plans and steps of the research should be mentioned. Also limitations of the research should be formulated by the authors.

Author Response

Response to Reviewer 1 Comments

Point 1

…it should be also explained which are the definitions of the aforementioned concepts used by authors further on in the research (e.g. own definition and delimitation of the concepts on which the present empirical research relies).

Response 1

Authors provided an additional information about concepts used in the research and the article:

  1. Employee motivation – lines 88-89
  2. Customer satisfaction and loyalty – lines 128-130
  3. Modelling approach – lines 163-165

Point 2
It is essential to provide an explanation regarding the time gap between the time period when the research was conducted (2013-2016, as stated in the text) and the submission date of this article. The empirical data is quite old (this is probably the main vulnerability of the article) and it should be emphasized its relevance in the current context.

Response 2

We have added the whole paragraph describing the consequences of the gap between the time of data collection and analysis  - Lines 254-262

Although the dataset may be considered quite old now, the analysis is aimed at the diagnosis of general characteristics of the relations occurring between researched employee and customer satisfaction (in the perspective of 3 year annual time period) rather than the specific industry or market related issues. Certainly, the picture of the market, companies and their customers evolve, but we assume that the postulated relationship is stable and time-independent in terms of its existence and general shape. In that sense the gap between the time of data collection and the article submission do not affect the overall research results and findings associated with the importance and the character of the “time gap” phenomenon occurring between the observed changes in the area of analyzed employee and customer processes.

Point 3

The discussion and conclusion part should be extended. The differences between the two sectors should be explained. It should be emphasized the practical relevance of the results for practitioners and managers from both industries and eventually recommendations for HR management and customer care management could be formulated. The research results, if properly explained and formulated could be very useful for companies.

Response 3

We added to the Discussion/Conclusions the following paragraphs:

(lines 507-519)

Including companies operating on two separate markets – banks and shopping centres, made it also possible to search and evaluate similarities and differences both in processes of creation of motivation, satisfaction and loyalty, as well as in the relationships connecting employee and customer behavior. The main findings from the comparative analysis of both categories of companies are:

  1. for both cases employee satisfaction has a significantly higher influence on customer satisfaction and loyalty than employee motivation;
  2. in the case of banks one may observe that the relation between employee motivation and customer areas is visibly stronger than the one observed within shopping centres;
  3. it was observed that shopping centres staff may have negative influence on the customer satisfaction and loyalty – when there are overmotivated, presumably by owners or managers, which is proven by negative correlation between changes of employee motivation and changes of customer satisfaction and loyalty level.

(lines 530-544)

Nowadays, both business practitioners as well as scientists seems to agree to a point that the main challenges of the 21st century business management will mostly concentrate on nothing else than managing old, well known intangible assets of the company (such as employee satisfaction & motivation or customer satisfaction & loyalty) by using the new, modern measurement and managing tools and techniques. Presented in the article the holistic model concept can be treated as a modern and innovative tool which can support decision making process of the company in the fields of HRM and CRM. This approach allowed managers to look at the both organization stakeholders (customers and employees) from the perspective of the one process. Thanks to it one can see how employees motivation and satisfaction affect customer satisfaction and loyalty (with reference to strength, direction, as well as time shift of that relation), what enables managers to better understand the vital part of the whole value creation process of the company. This knowledge can give a very precise information on how to efficiently and effectively allocate the organization resources in the fields of HRM and CRM. What is even more important this holistic PLS-SEM modeling approach gives managers ability to possess quantitative data of the both analyzed processes (which are strictly intangible in their nature) what makes them much easier to control and manage .

Point 4

If available, further plans and steps of the research should be mentioned. Also limitations of the research should be formulated by the authors.

Response 4

We added to the Discussion/Conclusions the following paragraphs (lines 545-567):

Despite the relevance and interesting findings of the presented study, there are several limitations that may diminish its validity. They are, mainly, related to the obtained sample. Although it is, at least structurally representative, random approach would be even more beneficial from the point of view of data quality – unfortunately, such approach, when it comes to shopping centres customers is, at least, very problematic. As a second main limitation we assume a long time delay between each stage of measurement. More frequent readouts, although difficult, would provide better data granularity and thus better insight into researched phenomena. There is also a question whether this kind of relationship, alongside with its strength, direction and a scale of the time shift, would be observed in different service sectors (e.g. car services, healthcare and wellbeing etc.). Finally, we assume that such research, due to cultural and social factors, maybe only valid for Europe. This limitation is often visible in different research related to customer satisfaction and loyalty - it is unknown whether the theory, constructs, measures, and relationships among constructs are applicable to other regions as well.

In order to avoid such limitations, we would recommend validating the presented theory on a smaller scale but with better sample and higher resolution – in a research conducted in particular company, involving more frequent – quarterly measurement and different approach to data gathering. It could involve, instead of quota (or even random) sampling, conducting the research on a relatively complete set of customers who would make a transaction in some limited period of time along with measuring, in very same time, employee levels of satisfaction and motivation. Another interesting way of extending and validating presented findings would be to replicate similar research on a company or companies from different markets. Moreover, for those who would like to implement such study in different geographical regions, it might be beneficial to evaluate the validity of assumed satisfaction, motivation and loyalty models.

Reviewer 2 Report

"Old wine in a new bootle"

Author Response

Nowadays, both business practitioners as well as scientists seems to agree to a point that the main challenges of the 21st century business management will mostly concentrate on nothing else than managing old well known intangible assets of the company (such as employee satisfaction & motivation or customer satisfaction & loyalty) by using the new, modern measurement and managing tools and techniques.

Reviewer 3 Report

Overall comment

This work describes and analyzes the existence of possible relationships between employee’s motivation (and satisfaction) and customer’s indices, in terms of satisfaction and loyalty as well.

For this purpose, two industries are analyzed (banking services and shopping centers), with the same studies, being conducted during three annual reference periods.

This study has been conducted by using the PLS-SEM method, regarding the analytical process. Based on the obtained results, the authors claim that there is a strong relationship between changes in the areas of employee and customer satisfaction  in the studied sectors, with a one-year time Shift.

The study also identifies that the strength of influence of the employee’s motivation level on  customers is clearly lower than the strength of influence of the employee satisfaction.

The study, allows to operationalize the discussed relationship in terms of strength, direction as well as  the time shift.

The paper seems to be well-organized, containing all the expected components, namely the   Introduction, Research Methods, Discussion and Analysis of results, and Conclusions.

The author’s results are convincing, given the purpose of the work. However, I still reluctant about the relation between the scope of this paper and the scope of the journal…what is the contribution of this paper in terms of sustainability or sustainable development?

Despite the development of the sections here (a section named “4.Discussion/Conclusions”?...why not just only “4.Conclusions”?. It seems to me that this section is more related with the conclusions of the work, rather than the discussion results). However, and in general, the authors have answered to the research question stated here.

Furthermore, the relevance of the subject is also high on the “present day”.

Some recommendations regarding this issue, can be found it below.

Some recommendations of improvement:

  Strong points:

  • Case studies and data used: The choice of the 2 case studies used here (3 shopping centers and 2 banks), are pertinent for the study carried out. Furthermore, the selection of these 2 scenarios, are well justified by the authors, since it represent two forms of activity in which the employee has direct contact with the customer, as well as the large number of staff employed by these 2 entities, which are used by a wide range of customers.

  • The relevance of the subject

  • Research methods used    

Weak points:

  • Literature review: most of the literature used here are old (2016 or less), which support the need of update, to reinforce the importance of this study

  • The relationship with sustainability or sustainable development: By considering that the authors have worked in a “social dimension” of sustainability… could this issue, be considered in a sustainable perspective? For me this is not clear…
  • The time period considered in this study: Although the period considered here, shows some degree of significance, the data has some years of existence, since we are in 2020 and the research was carried out in the years 2013-2016..
  • Discussion of results: Should be also improved and if it is possible, discussing the results based on other works regarding the methods which are being compared with the proposed one
  • Future work - Regarding the conclusions’ section, and despite the main research question, pointed and answered, based on the achieved results, the authors should better explore the “future work” in the same section, by pointing (for instance) some clues to a reader, who might want to pursue the research. One example, is regarding the deployment of the same models used here, into other sectors in order to better understand if the results achieved here are the same or not.

Author Response

Response to Reviewer 3 Comments

Point 1

Literature review: most of the literature used here are old (2016 or less), which support the need of update, to reinforce the importance of this study

Response 1

We extended the Introduction and Discussion (research limitations) parts of the article by referring to main findings and conclusions from six up-to-date research articles:

  1. Nielsen, J. D., Thompson, J. A., Wadsworth, L. L., Vallett, J. D. (2020), “The moderating role of calling in the work–family interface: Buffering and substitution effects on employee satisfaction”, Journal of Organizational Behavior, 41, Issue 7.
  2. Lorincová, S., Hitka, M., Bajzíková, L., Weberová, D. (2019), "Are the motivational preferences of employees working in small enterprises in Slovakia changing in time?", Entrepreneurship and Sustainability Issues, Vol. 6, Issue 4, pp. 1618-1635.
  3. Hitka, M., Rózsa, Z., Potkány, M., Ližbetinová, L. (2019), “Factors forming employee motivation influenced by regional and age-related differences”, Journal of Business Economics and Management, Vol. 20, Issue 4, pp. 674-693.
  4. Shokouhyar, S., Shokoohyar, S., Safari, S. (2020), “Research on the influence of after-sales service quality factors on customer satisfaction”, Journal of Retailing and Consumer Services, Vol. 56:102139.
  5. Murali, S., Pugazhendhi, S., Muralidharan, C. (2016), "Modelling and Investigating the relationship of after sales service quality with customer satisfaction, retention and loyalty“, Journal of Retailing and Consumer Services, Vol. 30(C), pp. 67-83.
  6. Yi, Y., and Nataraajan, R. (2018), “Customer satisfaction in Asia”, Psychology & Marketing, Vol. 35, Issue 6, pp. 387-391.

Point 2

The relationship with sustainability or sustainable development: By considering that the authors have worked in a “social dimension” of sustainability… could this issue, be considered in a sustainable perspective? For me this is not clear…

Response 2

The following paragraphs have been added to the introduction section [lines 52-66]:

The discussed research is, in general, related to the relationship between employee behaviour and its impact on customers (delayed) reactions. I such perspective it is directly connected with two dimensions of sustainability – social and economic. Assuming  an customer-centric sustainability as defined by Sheth et al., social dimension is related to impact of consumption on personal, while economic – on economic well-being. Customer satisfaction, and, in general, positive relation with company may and will affect both areas e.g. by reducing stress, creating positive social interactions and relationships, enabling more well-considered purchases, and thus reducing both costs for consumers, and impact on the environment.

From the companies’ point of view understanding the character of the relationship between employee and customers, especially in terms of time between stimuli and reactions may be even more important. Assumption that market’s reaction for possible changes in employee motivation or his or her satisfaction is not immediate, should lead to reduction of additional actions aimed at employee (e.g. trainings, reorganizations, but also layoffs) or directly at customers (e.g. marketing), therefore saving resources and labor, and, possibly, creating better work environment and enabling sustainable development of an organization.

Point 3

The time period considered in this study: Although the period considered here, shows some degree of significance, the data has some years of existence, since we are in 2020 and the research was carried out in the years 2013-2016..

Response 3

We have added the whole paragraph describing the consequences of the gap between the time of data collection and research conduction - Lines 254-262

Although the dataset may be considered quite old now, the analysis is aimed at the diagnosis of general characteristics of the relations occurring between researched employee and customer satisfaction (in the perspective of 3 year annual time period) rather than the specific industry or market related issues. Certainly, the picture of the market, companies and their customers evolve, but we assume that the postulated relationship is stable and time-independent in terms of its existence and general shape. In that sense the gap between the time of data collection and the article submission do not affect the overall research results and findings associated with the importance and the character of the “time gap” phenomenon occurring between the observed changes in the area of analyzed employee and customer processes.

Point 4

Discussion of results: Should be also improved and if it is possible, discussing the results based on other works regarding the methods which are being compared with the proposed one

Response 4

The following paragraphs has been added to the discussion/conclusion section:

(Lines 507-519)

Including companies operating on two separate markets – banks and shopping centres, made it also possible to search and evaluate similarities and differences both in processes of creation of motivation, satisfaction and loyalty, as well as in the relationships connecting employee and customer behavior. The main findings from the comparative analysis of both categories of companies are:

  1. for both cases employee satisfaction has a significantly higher influence on customer satisfaction and loyalty than employee motivation;
  2. in the case of banks one may observe that the relation between employee motivation and customer areas is visibly stronger than the one observed within shopping centres;
  3. it was observed that shopping centres staff may have negative influence on the customer satisfaction and loyalty – when there are overmotivated, presumably by owners or managers, which is proven by negative correlation between changes of employee motivation and changes of customer satisfaction and loyalty level.

(Lines 526-545)

 Its innovativeness and academic contribution is related to a completely new approach to evaluation the relationship between employee motivation and satisfaction, and customer satisfaction and loyalty, which include the existence of the time shift between them. Such a multi-stage research perspective is unique and currently not explored either by business or scientific research.

Nowadays, both business practitioners as well as scientists seems to agree to a point that the main challenges of the 21st century business management will mostly concentrate on nothing else than managing old, well known intangible assets of the company (such as employee satisfaction & motivation or customer satisfaction & loyalty) by using the new, modern measurement and managing tools and techniques. Presented in the article the holistic model concept can be treated as a modern and innovative tool which can support decision making process of the company in the fields of HRM and CRM. This approach allowed managers to look at the both organization stakeholders (customers and employees) from the perspective of the one process. Thanks to it one can see how employees motivation and satisfaction affect customer satisfaction and loyalty (with reference to strength, direction, as well as time shift of that relation), what enables managers to better understand the vital part of the whole value creation process of the company. This knowledge can give a very precise information on how to efficiently and effectively allocate the organization resources in the fields of HRM and CRM. What is even more important this holistic PLS-SEM modeling approach gives managers ability to possess quantitative data of the both analyzed processes (which are strictly intangible in their nature) what makes them much easier to control and manage.

Point 5

Future work - Regarding the conclusions’ section, and despite the main research question, pointed and answered, based on the achieved results, the authors should better explore the “future work” in the same section, by pointing (for instance) some clues to a reader, who might want to pursue the research. One example, is regarding the deployment of the same models used here, into other sectors in order to better understand if the results achieved here are the same or not.

Response 5

We have added the whole paragraph describing limitations of the study and possible further research - Lines 545-567

Despite the relevance and interesting findings of the presented study, there are several limitations that may diminish its validity. They are, mainly, related to the obtained sample. Although it is, at least structurally representative, random approach would be even more beneficial from the point of view of data quality – unfortunately, such approach, when it comes to shopping centres customers is, at least, very problematic. As a second main limitation we assume a long time delay between each stage of measurement. More frequent readouts, although difficult, would provide better data granularity and thus better insight into researched phenomena. There is also a question whether this kind of relationship, alongside with its strength, direction and a scale of the time shift, would be observed in different service sectors (e.g. car services, healthcare and wellbeing etc.). Finally, we assume that such research, due to cultural and social factors, maybe only valid for Europe. This limitation is often visible in different research related to customer satisfaction and loyalty - it is unknown whether the theory, constructs, measures, and relationships among constructs are applicable to other regions as well.

In order to avoid such limitations, we would recommend validating the presented theory on a smaller scale but with better sample and higher resolution – in a research conducted in particular company, involving more frequent – quarterly measurement and different approach to data gathering. It could involve, instead of quota (or even random) sampling, conducting the research on a relatively complete set of customers who would make a transaction in some limited period of time along with measuring, in very same time, employee levels of satisfaction and motivation. Another interesting way of extending and validating presented findings would be to replicate similar research on a company or companies from different markets. Moreover, for those who would like to implement such study in different geographical regions, it might be beneficial to evaluate the validity of assumed satisfaction, motivation and loyalty models.

Reviewer 4 Report

-  Authors should explain the academic contribution of the work developed.Highlighting what is innovative / original about the existing literature.

- Authors should develop the conclusions of the work and refer in more detail to the next steps of the work.

- Authors must correct the numbering of references throughout the text.
The first reference must be [1], the second to [2], and so on.

- Authors should explain better the figure 2.

- In the Introduction section, authors should describe the structure of the paper.

Author Response

Response to Reviewer 4 Comments

Point 1
Authors should explain the academic contribution of the work developed. Highlighting what is innovative / original about the existing literature.

Response 1

The following fragment has been added to the discussion/conclusion paragraph (lines 525-529):

Its innovativeness and academic contribution is related to a completely new approach to evaluation the relationship between employee motivation and satisfaction, and customer satisfaction and loyalty, which include the existence of the time shift between them. Such a multi-stage research perspective is unique and currently not explored either by business or scientific research.

Point 2

Authors should develop the conclusions of the work and refer in more detail to the next steps of the work.

Response 2

We added to the Discussion/Conclusions the following paragraphs:

(lines 507-519):

Including companies operating on two separate markets – banks and shopping centres, made it also possible to search and evaluate similarities and differences both in processes of creation of motivation, satisfaction and loyalty, as well as in the relationships connecting employee and customer behavior. The main findings from the comparative analysis of both categories of companies are:

  1. for both cases employee satisfaction has a significantly higher influence on customer satisfaction and loyalty than employee motivation;
  2. in the case of banks one may observe that the relation between employee motivation and customer areas is visibly stronger than the one observed within shopping centres;
  3. it was observed that shopping centres staff may have negative influence on the customer satisfaction and loyalty – when there are overmotivated, presumably by owners or managers, which is proven by negative correlation between changes of employee motivation and changes of customer satisfaction and loyalty level.

(lines 530-567):

Nowadays, both business practitioners as well as scientists seems to agree to a point that the main challenges of the 21st century business management will mostly concentrate on nothing else than managing old, well known intangible assets of the company (such as employee satisfaction & motivation or customer satisfaction & loyalty) by using the new, modern measurement and managing tools and techniques. Presented in the article the holistic model concept can be treated as a modern and innovative tool which can support decision making process of the company in the fields of HRM and CRM. This approach allowed managers to look at the both organization stakeholders (customers and employees) from the perspective of the one process. Thanks to it one can see how employees motivation and satisfaction affect customer satisfaction and loyalty (with reference to strength, direction, as well as time shift of that relation), what enables managers to better understand the vital part of the whole value creation process of the company. This knowledge can give a very precise information on how to efficiently and effectively allocate the organization resources in the fields of HRM and CRM. What is even more important this holistic PLS-SEM modeling approach gives managers ability to possess quantitative data of the both analyzed processes (which are strictly intangible in their nature) what makes them much easier to control and manage.

Despite the relevance and interesting findings of the presented study, there are several limitations that may diminish its validity. They are, mainly, related to the obtained sample. Although it is, at least structurally representative, random approach would be even more beneficial from the point of view of data quality – unfortunately, such approach, when it comes to shopping centres customers is, at least, very problematic. As a second main limitation we assume a long time delay between each stage of measurement. More frequent readouts, although difficult, would provide better data granularity and thus better insight into researched phenomena. There is also a question whether this kind of relationship, alongside with its strength, direction and a scale of the time shift, would be observed in different service sectors (e.g. car services, healthcare and wellbeing etc.). Finally, we assume that such research, due to cultural and social factors, maybe only valid for Europe. This limitation is often visible in different research related to customer satisfaction and loyalty - it is unknown whether the theory, constructs, measures, and relationships among constructs are applicable to other regions as well.

In order to avoid such limitations, we would recommend validating the presented theory on a smaller scale but with better sample and higher resolution – in a research conducted in particular company, involving more frequent – quarterly measurement and different approach to data gathering. It could involve, instead of quota (or even random) sampling, conducting the research on a relatively complete set of customers who would make a transaction in some limited period of time along with measuring, in very same time, employee levels of satisfaction and motivation. Another interesting way of extending and validating presented findings would be to replicate similar research on a company or companies from different markets. Moreover, for those who would like to implement such study in different geographical regions, it might be beneficial to evaluate the validity of assumed satisfaction, motivation and loyalty models.

Point 3

Authors must correct the numbering of references throughout the text.
The first reference must be [1], the second to [2], and so on.

Response 3

Edition issues – the numbering of the references have been changed.

Point 4

Authors should explain better the figure 2.

Response 4

We have added the whole paragraph describing in more details both Figure 1 and 2 - Lines 447-458

In the Figure 2 one can see separate statistical data obtained for customer satisfaction and loyalty as well as employee motivation and satisfaction (consequences of the specific customer model construction for banking services – see paragraph 2.2.3). In that case authors had obtained four different path relations results (employee motivation – customer satisfaction; employee satisfaction – customer satisfaction; employee motivation – customer loyalty; employee satisfaction – customer loyalty) for each “time gap” character (T2:T2-T1 / T3:T3-T2 time periods and T2-T1:T2-T1 / T3-T2:T3-T2 time periods). In case of Figure 1 the customer model construction for shopping centres combines areas of customer satisfaction and loyalty into one meta-area (see paragraph 2.2.2) and it results in obtaining of only two different path relations results (employee motivation – customer satisfaction & loyalty; employee satisfaction – customer satisfaction & loyalty) for each of the abovementioned “time gap” character.

Point 5

In the Introduction section, authors should describe the structure of the paper.

Response 5

We added to the introduction section the following paragraph (Lines 52-60):

In the article authors first concentrate on literature review concerning three main areas: modeling of employee motivation and satisfaction; modeling of customer satisfaction and the relationship between abovementioned phenomena. Then, the explanation and description of the research subject, research period and sampling procedure has been given. Afterwards authors in details explain the process of the construction of SEM models used in the research. Results of the research, which are the next part of the paper, are presented in the tables as well as in the graphical form of the two holistic models of relations between employee and customer processes for both analyzed industries (Figure 1 - shopping centres and Figure 2 – banks). The whole article is summarized by the conclusions, discussion and the research limitations.

Round 2

Reviewer 3 Report

Final comments:

Please, check the reference’s format according to the journal’s guidelines, as well as the remain part of the paper.